# Mandible Segmentation of Dental CBCT Scans Affected by Metal Artifacts Using Coarse-to-Fine Learning Model

**DOI:** 10.3390/jpm11060560

**Published:** 2021-06-16

**Authors:** Bingjiang Qiu, Hylke van der Wel, Joep Kraeima, Haye Hendrik Glas, Jiapan Guo, Ronald J. H. Borra, Max Johannes Hendrikus Witjes, Peter M. A. van Ooijen

**Affiliations:** 13D Lab, University Medical Center Groningen, University of Groningen, Hanzeplein 1, 9713 GZ Groningen, The Netherlands; b.qiu@umcg.nl (B.Q.); h.van.der.wel@umcg.nl (H.v.d.W.); j.kraeima@umcg.nl (J.K.); h.h.glas@umcg.nl (H.H.G.); m.j.h.witjes@umcg.nl (M.J.H.W.); 2Department of Radiation Oncology, University Medical Center Groningen, University of Groningen, Hanzeplein 1, 9713 GZ Groningen, The Netherlands; p.m.a.van.ooijen@umcg.nl; 3Data Science Center in Health (DASH), University Medical Center Groningen, University of Groningen, Hanzeplein 1, 9713 GZ Groningen, The Netherlands; 4Department of Oral and Maxillofacial Surgery, University Medical Center Groningen, University of Groningen, Hanzeplein 1, 9713 GZ Groningen, The Netherlands; 5Medical Imaging Center (MIC), University Medical Center Groningen, University of Groningen, Hanzeplein 1, 9713 GZ Groningen, The Netherlands; r.j.h.borra@umcg.nl

**Keywords:** mandible segmentation, cone-beam computed tomography (CBCT), computed tomography (CT), metal artifacts, 3D virtual surgical planning (3D VSP), convolutional neural networks

## Abstract

Accurate segmentation of the mandible from cone-beam computed tomography (CBCT) scans is an important step for building a personalized 3D digital mandible model for maxillofacial surgery and orthodontic treatment planning because of the low radiation dose and short scanning duration. CBCT images, however, exhibit lower contrast and higher levels of noise and artifacts due to extremely low radiation in comparison with the conventional computed tomography (CT), which makes automatic mandible segmentation from CBCT data challenging. In this work, we propose a novel coarse-to-fine segmentation framework based on 3D convolutional neural network and recurrent SegUnet for mandible segmentation in CBCT scans. Specifically, the mandible segmentation is decomposed into two stages: localization of the mandible-like region by rough segmentation and further accurate segmentation of the mandible details. The method was evaluated using a dental CBCT dataset. In addition, we evaluated the proposed method and compared it with state-of-the-art methods in two CT datasets. The experiments indicate that the proposed algorithm can provide more accurate and robust segmentation results for different imaging techniques in comparison with the state-of-the-art models with respect to these three datasets.

## 1. Introduction

Three-dimensional (3D) virtual surgical planning (VSP) technique is commonly used for orthodontic diagnosis, orthognathic diagnosis and surgery planning because it allows for pre- or post-operative simulation of surgical options [1]. Accurate mandible segmentation plays a critical role in the 3D VSP. 3D mandible surface models in 3D VSP are created and superimposed to demonstrate the orthodontic changes both visually and quantitatively (including pre- and post- operation). Cone-beam computed tomography (CBCT) is widely applied in 3D VSP because of its low radiation doses and short scanning duration. However, teeth, tooth fillings, and dental braces in orthodontic treatment and metal implants in orthognathic treatment are high attenuation materials which cause high noise and low contrast in visual impressions of CBCT images. Specifically, weak and false edges in parts of condyles and teeth often appear in the CBCT images. Furthermore, it is difficult to identify the boundaries of mandibles since the dental braces and metal implants negatively affect the image quality in CBCT, as shown in Figure 1. Therefore, it is challenging for orthodontic or orthognathic VSP to accurately perform mandibular segmentation in CBCT. Consequently, a large amount of manual work is required to reconstruct 3D mandible models. The patient-specific orthodontic or orthognathic treatment planning is restricted and delayed by this time-consuming procedure.

To reduce the workload of mandible segmentation, a number of traditional segmentation methods have been developed in the past, including statistical shape model [2] as well as machine learning methods [3,4,5,6]. Sebastian et al. [2] presented a statistical shape model (SSM) based mandible segmentation approach. They introduced an optimized correspondence to their SSM model. Wang et al. [3] employed a majority voting method and combined it with random forest for mandible segmentation. Rarasmaya et al. [4] proposed a method based on histogram thresholding and polynomial fitting to segment mandibular cortical bone in CBCT scans. Oscar et al. [5] used super-voxels and graph clustering for mandible segmentation in CBCT images. Fan et al. [6] proposed an automatic approach for segmenting mandibles from CBCT using a marker-based watershed transform. However, some of these traditional techniques require mandible shape prior to initialization, and the performances of these methods are often affected by noise or metal artifacts. Furthermore, it is difficult to adjust the model parameters according to the overall characteristics of the target contour [7].

With the development of convolutional neural networks (CNN), many approaches have introduced the CNN for mandible segmentation. Ibragimo et al. [8] presented the first attempt of using the deep learning concept of CNN to segment organs at risk (OARs) in head and neck CT scans. The AnatomyNet [9] is built upon the popular 3D Unet architecture using residual blocks in encoding layers and a new loss function combining Dice score and focal loss in the training process. A fully CNN (FCNN) method with a shape representation model for segmentation of organs at risk in CT scans was presented in [10]. Qiu et al. [11] developed a novel technique, RSegUnet, for mandible segmentation in conventional CT scans. This kind of network architecture combines the recurrent unit and the normal segmentation network. RSegUnet has been proven able to accurately segment the mandible parts with weak boundaries, such as condyles and ramus, since the network considers the continuity of neighborhood slices for the scans [11]. The recurrent segmentation network relies on the spatial connections between pixels of the mandible. However, this approach is vulnerable to spatial discontinuities such as metal artifacts in the tooth region and can suffer from oversegmentation once the upper and lower teeth are connected in the scans. Although these methods have led to some performance improvements, they still exhibit some disadvantages such as missing parts of the ramus when metal artifacts are present.

As mentioned in the literature, automatic mandible segmentation is still far from a solved problem, especially for CBCT scans, and post-processing for the mandible often requires significant manual interaction to achieve useful results for clinical practice. In this work, we aim to develop an accurate mandible segmentation algorithm to overcome the inaccurate prediction for 3D VSP in CBCT.

Motivated by the concept of curriculum learning [12], we propose a novel mandible segmentation approach based on a coarse-to-fine learning framework to solve the aforementioned challenges in mandible segmentation of CBCT. Curriculum learning draws upon a learning idea that follows a learning order from easy to difficult tasks [12]. Specifically, a complex task can be solved by dividing it into simple sub-tasks, then starting from the simplest sub-task and progressing to the more difficult sub-tasks. In this study, we propose a hybrid method which consists of a coarse stage and fine stage, in which the coarse stage makes use of 3D CNN for predicting the mandible-like organ and the fine stage utilizes the recurrent segmentation CNN for fine mandible segmentation in CBCT images which are mostly affected by metal artifacts. The proposed approach aims at overcoming the oversegmentation in some parts of the tooth regions and undersegmentation in the weak edges of the ramus and condyles. The coarse segmentation from the coarse stage guides the segmentation of the mandible in the fine stage, and therefore decreases the difficulty of segmenting the mandible from CBCT. Along with this coarse-to-fine segmentation (named as C2FSeg) task, we design a network by stacking a 3D SegUnet and a recurrent SegUnet. In addition, we extend the proposed segmentation network with a hybrid loss proposed by Taghanaki et al. [13], which has been demonstrated to offer superior performance in many visual applications.

This paper proposes a novel mandible segmentation approach for artifact-affected CBCT/CT with two main contributions:First, we apply the concept of curriculum learning to split the mandible segmentation into two sub-tasks. We extract the mandible-like organ using a 3D Unet in the coarse stage and then apply the mandible-like organ into the recurrent segmentation network in the fine stage. In comparison with other CNN approaches, the proposed segmentation approach is robust against metal artifacts.Second, the proposed model achieves promising performance on the dataset of CBCT scans of dental braces. Furthermore, the proposed model achieves a promising performance on the conventional CT dataset and Public Domain Database of the Computational Anatomy (PDDCA) dataset.

## 2. Methodology

From the perspective of the framework, a coarse-to-fine mandible segmentation approach (C2FSeg) is proposed according to curriculum learning [12]. The C2FSeg consists of two main components: coarse stage and fine stage, in which the coarse stage obtains the mandible-like organs, while the fine stage reduces the false positive rate by embedding the overall information of the mandible-like organs and the neighboring information. In the coarse stage, potential mandible candidates are first identified, and then reduction of the false positives (FPs) within the candidates is performed in the fine stage. The coarse model identifies potential mandible candidates, and the fine model reduces the false positives (FPs) within the candidates. The overview of the mandible segmentation framework is given in Figure 2.

### 2.1. Curriculum Learning in Mandible Segmentation

Curriculum learning describes a type of learning in which tasks can start with simple tasks before the number of difficult tasks is gradually increased. This learning method is proposed by [12]. It assumes that the curriculum learning can improve the convergence speed of the training process and find a better local minimum [12]. To elaborate upon the proposed C2FSeg approach, we first formulate a segmentation model that can be generalized to both coarse stage and fine stage; we will customize the segmentation model to these two stages in Section 2.2 and Section 2.3, respectively.

Let $X=\{x^{1},\ldots ,x^{t},\ldots ,x^{n}\}$ be the head and neck scan volume, where *X* belongs to the CBCT image domain, $X\in \Omega =R^{n\times w\times h}$, where *n*, *w* and *h* represent slice number, width and height, respectively. The corresponding ground truth is $Y=\left\{y^{1},\ldots ,y^{t},\ldots ,y^{n}\right\}\in {\left\{0,1\right\}}^{n\times w\times h}$, where *t* denotes the *t*-th slice of the CT scan. Let $\hat{Y}=\left\{{\hat{y}}^{1},\ldots ,{\hat{y}}^{t},\ldots ,{\hat{y}}^{n}\right\}\in S\in {\left[0,1\right]}^{n\times w\times h}$ denote the predicted segmentation (Ω→S). We denote a segmentation task by an operator *F*, i.e., $\hat{Y}=F\left(X,\theta \right)$, where θ indicates model parameters. Specifically, in a CNN model with *L* layers and parameters $\theta =\{w^{1},w^{2},\ldots ,w^{L};b^{1},b^{2},\ldots ,b^{L}\}$, $\left\{w^{1},w^{2},\ldots ,w^{L}\right\}$ is a set of weights and $\left\{b^{1},b^{2},\ldots ,b^{L}\right\}$ is a set of biases. According to the concept of curriculum learning, in which a task can be divided into several simple sub-tasks, the task is defined as $F=F_{1},\ldots ,F_{s}$, where *s* represents the number of sub-tasks. Therefore, the predicted segmentation $\hat{Y}=F\left(X,\theta \right)$ can be rewritten as $\hat{Y}=F_{s}\left(F_{s-1},\ldots ,F_{2}\left(F_{1}\left(X,\theta_{1}\right),X,\theta_{2}\right),\ldots ,X,\theta_{s-1}),X,\theta_{s}\right)$. Although this structure can improve the performance of the task and reduce the difficulty of the task, this model exponentially increases required computing resources, resulting in low efficiency. A small *s* is sufficient to handle the problem of mandible segmentation. We use s=2 in this study, i.e.,
(1)$$\hat{Y}=F_{2}\left(F_{1}\left(X,\theta_{1}\right),X,\theta_{2}\right),$$
where $F_{1}$ and $F_{2}$ denote the models from coarse stage and fine stage, respectively.

### 2.2. Coarse Stage: Mandible-Like Organ Prediction

One of the obstacles to training 3D deep networks is the problem of “insufficient memory”. A common solution is to train a 3D CNN from smaller sub-volumes (3D patches) and test it by sliding window [14]: that is, to perform 3D segmentation on densely and uniformly sampled sub-volumes. In the coarse stage, the input of the coarse stage is cropped from the whole CBCT scan volume *X* denoted by $X^{c}$, $X^{c}=Crop\left(X\right)\in sub\left(\Omega \right)=R^{n^{c}\times w^{c}\times h^{c}},\left(n^{c}\leq n,w^{c}\leq w,h^{c}\leq h\right)$, where $n^{c}$, $w^{c}$ and $h^{c}$ represent the depth, width and height of the cropped volume, respectively. The coarse segmentation model can be formulated as:(2)$${\hat{Y}}_{1}^{c}=F_{1}\left(X^{c},\theta_{1}\right).$$

The goal of this stage is to efficiently produce the rough mandible segmentations ${\hat{Y}}_{1}^{c}$ from the complex background, which can remove regions that are segmented as non-mandible with high confidence in order to obtain an approximate mandible volume. To be used in the following fine stage, we need to map the sub-volume predictions ${\hat{Y}}_{1}^{c}$ back to exactly the same location given by $X^{c}=Crop\left(X\right)$ after all the positions are traversed. The mathematical definition is ${\hat{Y}}_{1}=UnCrop\left({\hat{Y}}_{1}^{c}\right)$.

As illustrated in Figure 3a, the coarse stage of our proposed C2FSeg network for capturing the mandible-like candidates is based on the 3D SegUnet, which is the original SegUnet [15] expanded from 2D to 3D. The 3D SegUnet consists of an encoder and a decoder, each of which has four convolutional blocks follwed by 3D maxpooling or 3D uppooling layers. The 3D convolutional block includes two convolution operations with a kernel of 3×3×3, each of which is followed by a batch normalization [16] and a rectified linear unit (ReLU) [17] activation function. The number of filters in the encoder starts at 32 and increases by a factor of 2 after every 3D convolutional block, while it declines by a factor of 2 in the decoder path. The number of feature maps is listed on the left of each convolutional block, and the convolutional layers are represented in Figure 3. In addition to the encoder and the decoder, we also use a cross connection to bridge the short-cut connection between the low-level and high-level layers, and we use a cross connection to transfer coordinates from maxpooling to uppooling. In the forward phase, the low-level feature maps extracted from the encoder are directly concatenated to the high-level feature maps, which can improve fine-scaled segmentation [18]. As for the backward phase, the high-level feature maps can be propagated backward through the connections. This approach can prevent the network from enduring gradient vanishing, which will hinder the convergence of the network in the training process [19]. The output of the 3D SegUnet is obtained by applying a convolutional layer with a kernel of 1×1×1 followed by the sigmoid function.

### 2.3. Fine Stage: False Positive Reduction

In the fine stage as shown in Figure 2b, a recurrent SegUnet (RSegUnet) is utilized to predict the segmentation map. SegUnet [15] is used as a basic element in the recurrent network. The network setting is the same as 3D SegUnet on the coarse stage and is performed by using a 2D kernel instead of using a 3D kernel, as illustrated in Figure 3b. RSegUnet adopts the structure of the recurrent neural network, which forms a directed acyclic graph, so that the recurrent connection between adjacent nodes can maintain its connectivity. Furthermore, RSegUnet can further learn the shape of the mandible based on its anatomical connectivity by using the spatial information from neighboring predictions. The coarse network spotted the potential mandible candidates, and we further refine the segmentation results by reducing the FPs of the coarse predictions.

To further utilize the information obtained from the prediction of the previous neighborhood slice, we use RSegUnet [11] to accurately segment the mandible in the fine stage. The sequential design of RSegUnet allows the network to learn anatomical structure continuity in 3D form. In the fine stage, RSegUnet, $F_{2}$, processes each slice sequentially. The input of RSegUnet is sampled from the scan volume *X* and the rough predictions from ${\hat{Y}}_{1}$ of the coarse stage. Here, $\hat{Y_{2}}=\left\{{\hat{y}}_{2}^{1},\ldots ,{\hat{y}}_{2}^{t},\ldots ,{\hat{y}}_{2}^{n}\right\}\in S_{2}\in {\left\{0,1\right\}}^{n\times w\times h}$ is the binary segmentation map generated from the fine stage ($\left\{\Omega ,S_{1}\right\}\to S_{2}$). In general, ${\hat{Y}}_{2}=F_{2}\left(\hat{Y_{1}},X,\theta_{2}\right)$. In this task, RSegUnet maps a sequence input slice ($x^{t},{\hat{y}}_{1}^{t},{\hat{y}}_{2}^{t-1}$) to a sequence output ${\hat{y}}_{2}^{t}$ of the same length, i.e., the output of the unfolded RSegUnet after *t* steps is represented as:(3)$${\hat{y}}_{2}^{t}=F_{2}\left({\hat{y}}_{2}^{\left(t-1\right)},x^{t},{\hat{y}}_{1}^{t}\right).$$

All in all, the fine stage of the proposed C2FSeg framework is illustrated in Figure 3b.

### 2.4. Loss

These two stages are trained separately using the same unified loss function. The loss function of each stage is a combination of Dice and binary cross entropy (BCE) loss. These loss functions are selected due to their potential to deal with imbalanced data.
(4)$$L=\omega_{1}\times L_{BCE}+\omega_{2}\times L_{Dice},$$
where $\omega_{1}$ and $\omega_{2}$ are the hyperparameters which adjust the amounts of BCE and Dice contribution in the loss function L. $L_{BCE}$ and $L_{Dice}$ are defined as follows:(5)$$L_{BCE}\left(\hat{y},y\right)=-\frac{1}{N}\sum_{i=1}^{N}y_{i}\log\left({\hat{y}}_{i}\right)+\left(1-y_{i}\right)\log\left(1-{\hat{y}}_{i}\right),$$
(6)$$L_{Dice}\left(\hat{y},y\right)=1-\frac{2\sum_{i=1}^{N}y_{i}{\hat{y}}_{i}}{\sum_{i=1}^{N}y_{i}+{\hat{y}}_{i}}.$$

Here, $y_{i}$ and ${\hat{y}}_{i}$ represent the ground truth and the predicted probability of pixel *i*, respectively, and *N* is the number of pixels.

According to Equation (Equation 5), the term $\left(1-y_{i}\right)\log\left(1-{\hat{y}}_{i}\right)$ penalizes false positives (FPs), as it is 0 when the prediction probability is correct, and $y_{i}\log\left({\hat{y}}_{i}\right)$ penalizes false negatives (FNs) [13]. Therefore, the BCE term is able to control the trade-off between FPs and FNs in the pixelwise segmentation task. In spite of that, the networks with only BCE as loss function are often prone to generate more false positives in the segmentation [20]. The study from [21] has proven that Dice loss yields better performance for one-target segmentation and is able to predict the fine appearance features of the object. Dice loss is based on the Dice coefficient metric, which measures the proportion of overlap between the resulting segmentation and the ground truth. Thus, the combination of loss functions can control the penalization of both FPs and FNs by the BCE term and simultaneously boost the model parameters out of local minima via Dice term.

The training procedure of RSegCNN is the same as that of the traditional CNN, where the trainable weights are updated with the backpropagation through time (BPTT) algorithm [22]. According to Equation (Equation 4), the loss for the *t*-th step with prediction $\hat{y}$ with respect to ground truth *y* is:(7)$$L^{t}\left({\hat{y}}^{t},y^{t}\right)=\omega_{1}\times L_{BCE}\left({\hat{y}}^{t},y^{t}\right)+\omega_{2}\times L_{Dice}\left({\hat{y}}^{t},y^{t}\right),$$
in which each $L^{t}$ is used only at step *t*.

The gradient $\frac{\partial L^{t}}{\partial {\hat{y}}_{j}^{t}}$ of the outputs at step *t*, for all j,t, is as follows:(8)$$\begin{array}{cc} \frac{\partial L^{t}}{\partial {\hat{y}}_{j}^{t}}= & \omega_{1}\frac{\partial L_{BCE}\left({\hat{y}}^{t},y^{t}\right)}{\partial {\hat{y}}_{j}^{t}}+\omega_{2}\frac{\partial L_{Dice}\left({\hat{y}}^{t},y^{t}\right)}{\partial {\hat{y}}_{j}^{t}}\\ = & -\frac{\omega_{1}}{N}\left(\frac{y_{j}^{t}}{{\hat{y}}_{j}^{t}}-\frac{1-y_{j}^{t}}{1-{\hat{y}}_{j}^{t}}\right)-\omega_{2}\frac{2y_{j}^{t}}{\sum_{i=1}^{N}y_{i}^{t}+{\hat{y}}_{i}^{t}}+\\ \; & \omega_{2}\frac{2\sum_{i=1}^{N}y_{i}^{t}{\hat{y}}_{i}^{t}}{{\left(\sum_{i=1}^{N}y_{i}^{t}+{\hat{y}}_{i}^{t}\right)}^{2}} \end{array}$$

### 2.5. Evaluation Metrics

For quantitative analysis of the experimental results, four performance metrics are used, including Dice coefficient (Dice), average symmetric surface distance (ASD) and 95% Hausdorff distance (95HD).

Dice coefficient is widely used to assess the performance of image segmentation algorithms [23]. It is defined as:(9)$$\mathrm{Dice}=\frac{2\sum_{i=1}^{N}y_{i}{\hat{y}}_{i}}{\sum_{i=1}^{N}y_{i}+{\hat{y}}_{i}}.$$

The average symmetric surface distance (ASD) is a measure for computing the average distance between the boundaries of two object regions [10]. It is defined as:(10)$$\mathrm{ASD}\left(A,B\right)=\frac{d(A,B)+d(B,A)}{2},$$
(11)$$d\left(A,B\right)=\frac{1}{N}\sum_{a\in A}\min_{b\in B}∥a-b∥,$$
where ∥.∥ represents the $L_{2}$ norm. *a* and *b* are corresponding points on the boundary of *A* and *B*.

Hausdorff distance (HD) measures the maximum distance of a point in a set *A* to the nearest point in the other set *B* [24]. It is defined as:(12)HD(A,B)=max(h(A,B),h(B,A))
(13)$$h\left(A,B\right)=\max_{a\in A}\min_{b\in B}∥a-b∥$$
where h(A,B) denotes the directed HD. The maximum HD is sensitive to contours. When the image is contaminated by noise or occluded, the original Hausdorff distance is prone to mismatch [25,26]. Thus, Huttenlocher proposed the concept of partial Hausdorff distance in 1933 [24]. The 95HD metric is similar to maximum HD. In brief, 95HD selects 95% of the closest points in set *B* to the point in set *A* in Equation (Equation 13) to calculate h(A,B):(14)$$95\mathrm{HD}=\max(h^{95\%}\left(A,B\right),h^{95\%}\left(B,A\right))$$
(15)$$h^{95\%}\left(A,B\right)=\max_{a\in A}\min_{b\in B^{95\%}}∥a-b∥$$

The purpose of using 95HD is to reduce the impact of a small subset of inaccurate prediction outliers on the overall assessment of segmentation quality.

## 3. Experiments

We evaluate our method on three datasets and compare our performance with state-of-the-art methods.

### 3.1. Datasets

#### 3.1.1. CBCT Dataset

A total of 59 orthodontic CBCT scans that had been heavily affected by metal artifacts were used in this study. All the CBCT scans were obtained on a Vatech PaXZenith3D (or Planmeca promax). Each scan consists of 431 to 944 slices with size of 992×992 to 495×495 pixels. The pixel spacing varies from 0.2 to 0.4 mm and the slice thickness varies from 0.2 to 0.4 mm. Of these CBCT scans, 38 are used for training, 1 is used for validation and 20 are used for testing. To train a CNN for bone segmentation in these CBCT scans, gold standard segmentation labels were required. These gold standard labels were created by the manual segmentation of all CBCT scans by three experienced medical engineers using Mimics software 20.0 (Materialise, Leuven, Belgium).

#### 3.1.2. CT Dataset

In addition, we also compare the proposed method with several state-of-the-art methods on two CT datasets. The collection of the patient datasets for medical research purposes was approved by the local medical ethical committee. The dataset contains 109 CT scans reconstructed with a kernel of Br64, I70h(s) or B70s. Each scan consists of 221 to 955 slices with size of 512×512 pixels. We randomly choose 52 cases as training, 8 cases as validation and 49 cases as test. The images have axial dimensions of 512 by 512 with slice numbers varying from 221 to 955. The pixel spacing varies from 0.35 to 0.66 mm, and the slice thickness varies from 0.6 to 0.75 mm. The manual mandible segmentation was performed using Mimics software version 20.0 (Materialise, Leuven, Belgium) by a trained researcher and confirmed by a clinician.

We also test the proposed strategy on the public dataset PDDCA [25]. This dataset contains 48 patient CT scans from the Radiation Therapy Oncology Group (RTOG) 0522 study, a multi-institutional clinical trial, together with manual segmentation of left and right parotid glands, brainstem, optic chiasm and mandible. Each scan consists of 76 to 360 slices with size of 512×512 pixels. The pixel spacing varies from 0.76 to 1.27 mm, and the slice thickness varies from 1.25 to 3.0 mm. According to the Challenge description, we follow the same training and testing protocol [25]. Forty of the 48 patients in PDDCA with manual mandible annotations are used in this study [25,27], in which the dataset is split into the training and test subsets, each with 25 (0522c0001-0522c0328) and 15 (0522c0555-0522c0878) cases, respectively [25].

### 3.2. Implementation Details

We implement all the experiments based on the PyTorch [28] platform developed by Facebook. The experiments are trained on a workstation equipped with an Nvidia P6000 or Tesla V100 GPU. For the data pre-processing, we simply truncated the raw intensity values to be within [−1000,2000], and then normalized each raw CT case to [0,1] to decrease the data variance caused by physical considerations of the medical device. Note that different CBCT/CT cases have different physical resolutions. As described in Section 3.1, we maintain their resolutions in a unified resolution of 512×512. The weights of the BCE loss term $\omega_{1}$ and the Dice loss term $\omega_{2}$ in the loss function are both set to 0.5. We use Adam optimization with a learning rate of $r=10^{-4}$.

For the coarse stage, we randomly sampled $n^{c}\times w^{c}\times h^{c}=64\times 128\times 128$ sub-volumes from the whole CT scan in the training phase. In this case, a sub-volume can either cover a portion of mandible voxels or be cropped from regions with non-mandible voxels, which acts as a hard negative mining to reduce the false positives. In the testing phase, a sliding window is carried out for the entire CT volume with a coarse step size that has small overlaps within each neighboring sub-volume. Specifically, for a testing volume with a size of (64,128,128), we have a total number of sub-volumes to be fed into the network and then combined to obtain the final prediction. For the fine stage, we sequentially sample the slices from the medical scan, the coarse predictions from the coarse stage and apply the mask from the previous unit.

### 3.3. Results

#### 3.3.1. Experiments on the CBCT Dataset

We compare our methods with numerous standard segmentation architectures such as Unet [18], Segnet [29], SegUnet [15], AttUnet [30] and RSegUnet [11]. Table 1 shows the performance comparison as well as the corresponding standard deviation for the mandible segmentation. The average Dice, $D_{ASD}$ and $D_{95HD}$ values of the proposed method are 95.31%, 1.2827 mm and 3.1258 mm, respectively. From the Table 1, it can be observed that the proposed method outperforms the existing approaches with respect to Dice, $D_{ASD}$ and $D_{95HD}$. These experimental results indicate that our proposed model with the C2FSeg learning method performs significantly better and achieves the highest overall Dice scores compared to other segmentation methods. According to Table 1, the proposed method also outperforms most other methods, with the second-lowest ASD and 95HD scores.

To better demonstrate the performance of the presented approach, several 2D and 3D view examples of the different algorithms are depicted in Figure 4 and Figure 5. Figure 4 shows some examples of ground truth (GT), Unet [18], Segnet [29], SegUnet [15], AttUnet [30], RSegUnet [11] and the proposed method. As shown in the first two rows of Figure 4, the other methods fail to obtain satisfactory results for the main mandible body, while the results from our proposed approach are much better. The third row in Figure 4 show that the proposed algorithm achieves better performances when the upper jaw teeth and lower jaw teeth appear within the same slice. The final row in Figure 4 illustrates that the proposed method can deal with the ambiguity and blurred boundaries common to CBCT scans of the condyles area.

Figure 5 illustrates two 3D view examples (the 1st, and 3rd rows) with the corresponding post-processed examples (the 2nd, and 4th rows) obtained from Unet [18], Segnet [29], SegUnet [15], AttUnet [30], RSegUnet [11] and the proposed method, respectively. The first case shown in Figure 5 demonstrate that the proposed method can effectively segment the angle area of the mandible. The second examples shown in Figure 5 show that the ramus, the coronoid process area and the teeth are missed by the other methods while the proposed method can tackle the thin parts of the mandible, which are almost always challenging mandible segmentation tasks.

Table 1, Figure 4 and Figure 5 indicate that the proposed approach is quite accurate in segmenting mandibles affected by metal artifacts in CBCT.

#### 3.3.2. Experiments on the CT Dataset

We also test the proposed method on a CT dataset. To quantitatively compare the proposed approach with other methods, we compute the Dice scores, ASD and 95HD values of the five methods. Table 2 lists the average Dice, ASD and 95HD, as well as the corresponding standard deviation. In general, the average values of these metrics obtained from our proposed method are better than those of the other methods. As shown in Table 2, it can be observed that our method yields the highest mean Dice score and the smallest mean errors in 95HD.

Figure 6 shows some examples from the original CT slice, the ground truth (GT) and the results obtained from Unet, Segnet, SegUnet, AttUnet, RSegUnet, and the proposed method. As shown in the first two rows of Figure 6, the other methods fail to obtain satisfactory results for the main mandible body and some parts of teeth, while the results from our proposed approach are much better. The three row in Figure 6 shows that the proposed algorithm achieves better performances when the upper jaw teeth and lower jaw teeth appear within the same slice. The final row in Figure 6 illustrates that the proposed method can process the condyles area, which is often ambiguous due to blurred boundaries in CT scans.

Moreover, two cases of the automatic segmentation (the 1st, and 3rd rows) and the corresponding post-processed examples (the 2nd, and 4th rows) in the 3D views of Unet, Segnet, SegUnet, AttUnet, RSegUnet and the proposed method are displayed in Figure 7. The first case shown in Figure 7 indicate that the proposed method can effectively segment the ramus area and body of the mandible. The second examples shown in Figure 7 show that the angle area and the teeth in the case are missed by the other methods, while the proposed method can tackle the thin parts of the mandible, which are almost always challenging mandible segmentation tasks. The conventional methods usually lead to erroneous segmentation within the whole mandible, as shown in Figure 7. The visual comparison of the automatic segmentation with manual segmentation demonstrates the effectiveness of our method with respect to automatic mandible segmentation. To summarize, Figure 6 and Figure 7 and Table 2 indicate that the proposed approach also works well with respect to the other datasets.

To further investigate the proposed method, it is preferable to test it with respect to a public dataset and measure the performance of the proposed method on the dataset. Here, we compare our proposed method with several state-of-the-art methods with respect to the PDDCA dataset. Table 3 also lists Dice, ASD and 95HD used in the Challenge paper [10,25]. According to Table 3, the performance of the proposed model surpasses the majority of the other methods. The proposed method outperforms other methods, with the third-highest mean Dice score, the lowest ASD and the lowest 95HD. For Dice score results, the segmentation result of our method is only slightly worse than RSegUnet [10], while it is better than RSegUnet in terms of ASD and 95HD.

## 4. Discussion

In this paper, we present a novel C2FSeg method for mandible segmentation that utilizes the curriculum learning strategy [12] of separating the task into two simpler sub-tasks (coarse-to-fine). We apply 3D SegUnet to look for mandible-like organs in the coarse stage. In the fine stage, recurrent SegUnet is then employed to finely segment the mandible based on the results from the coarse stage. Quantitative evaluation results shown in Table 1, Table 2 and Table 3 demonstrate that our proposed approach outperforms the state-of-the-art methods for mandible segmentation. In addition, qualitative visual inspection in Figure 4, Figure 5, Figure 6 and Figure 7 illustrates that our automatic segmentation approach performs quite well in comparison with the ground truth. The direct comparison in PDDCA, as shown in Table 3, illustrates that the proposed method significantly improves mandible segmentation. Remarkably, we found that this segmentation architecture is very robust for weak and blurry edge segmentation. For instance, the networks can segment both condyles and ramus of the mandible quite well, even under the influence of strong metal artifacts. In addition, the results based on the two CT datasets indicate that our proposed approach offers excellent generalization ability, since the images in the two datasets were produced by different imaging technologies.

This method takes advantage of the concept of curriculum learning to simplify the difficult task to several easy sub-tasks. Therefore, the proposed approach can help to learn the rough mandible structure in the coarse stage. Furthermore, the proposed approach utilizes a recurrent network in the fine stage to extract spatial information of objects based on mandible-like candidates from the first stage. The experimental results show that the proposed approach is feasible and effective in 3D mandible segmentation and that it can also be applied to other segmentation tasks. This method can support further research on the 3D image segmentation. It can also help overcome the disadvantage of cropping volume for the 3D network due to the high memory consumption, as well as accomplish 3D segmentation tasks. Furthermore, in this study, 3D SegUnet is used for searching mandible-like organs, and many other networks which have the similar ability for seeking mandible candidates can be used to replace the 3D SegUnet in the coarse stage.

Despite the promising results, there are a few limitations in this study. First, in the experiment, we use 59 orthodontic CBCT scans, 109 CT scans and a public dataset (PDDCA) for the training and the validation of the proposed approach. This is because the collection of CBCT scans is limited. For future work, we will focus on the validation of the C2FSeg approach to experiment on a large number of CBCT scans in order to prove the feasibility of the approach in the clinical setting of 3D VSP. Second, the proposed C2FSeg is a two-stage approach in which the two stages are trained separately. This increases the training duration of the model. In the future, we also aim at improving the efficiency of the approach.

## 5. Conclusions

In this paper, we attempt to address the problem of mandible segmentation using the coarse-to-fine approach. The coarse stage of our algorithm attempts to obtain probable mandible-like candidates in 3D volume. The fine stage of our approach attempts to reduce the false positives detected from the estimated mandible-like organs. First, we employ a patch-based mandible detector, wherein scans are divided into overlapping patches which are classified as mandible or non-mandible. Second, we utilize the recurrent CNN to finely segment the mandible following the coarse stage. The proposed algorithm is evaluated on three datasets: an orthodontic CBCT dataset polluted by metal artifacts, a CT dataset and a PDDCA dataset. Experimental results show that our method can achieve high accuracy for mandible segmentation in comparison with ground truth. The method overcomes the problem of weak mandible boundaries caused by low radiation and strong metal artifacts.

## Figures and Tables

**Figure 1 jpm-11-00560-f001:**
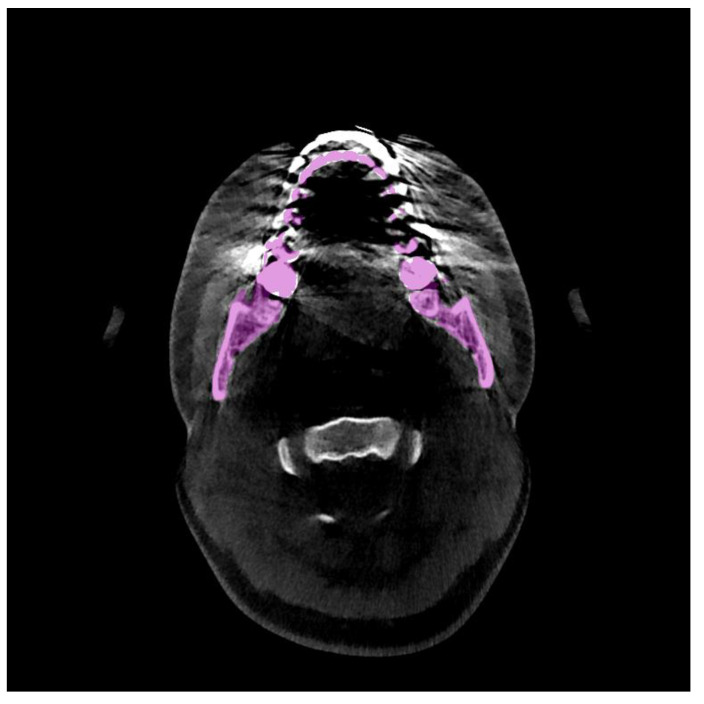
Example of manual annotation of the mandible in a CBCT image with strong metal artifacts.

**Figure 2 jpm-11-00560-f002:**
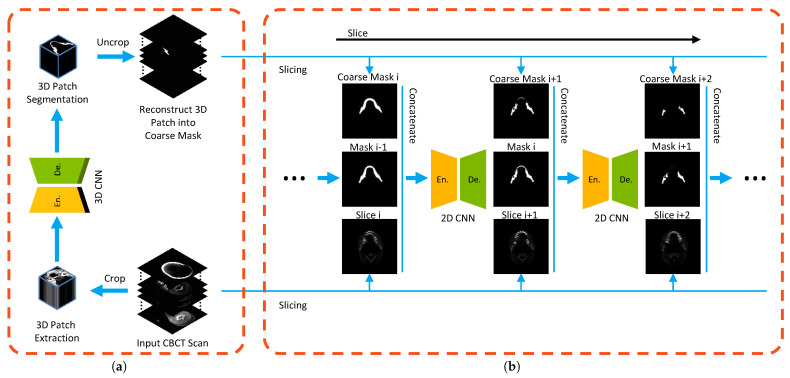
Overview of the proposed method consisting of (**a**) a 3D CNN for rough mandible segmentation and (**b**) a 2D recurrent segmentation network for further accurate mandible segmentation. The implementation of stage (**a**) is as follows: input scan → 3D patch extraction → 3D CNN (3D SegUnet based) → 3D patch segmentation → reconstruct the 3D patches into 3D segmentation of the scan. (**b**) The implementation of stage (**b**) is as follows: fusion with the coarse mask, the output probability maps and input data → RSegCNN (recurrent SegUnet) → accurate mandible prediction.

**Figure 3 jpm-11-00560-f003:**
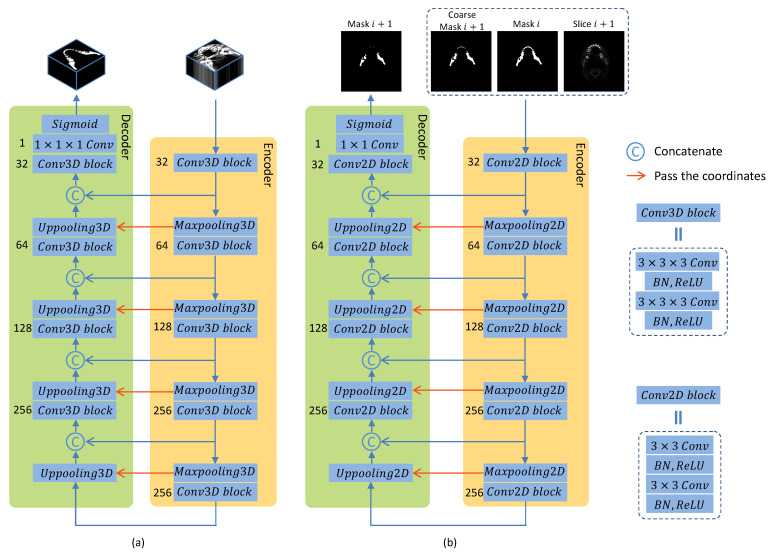
The detailed architecture setting of the proposed method consists of a 3D CNN (**a**) and a 2D recurrent segmentation network (**b**).

**Figure 4 jpm-11-00560-f004:**
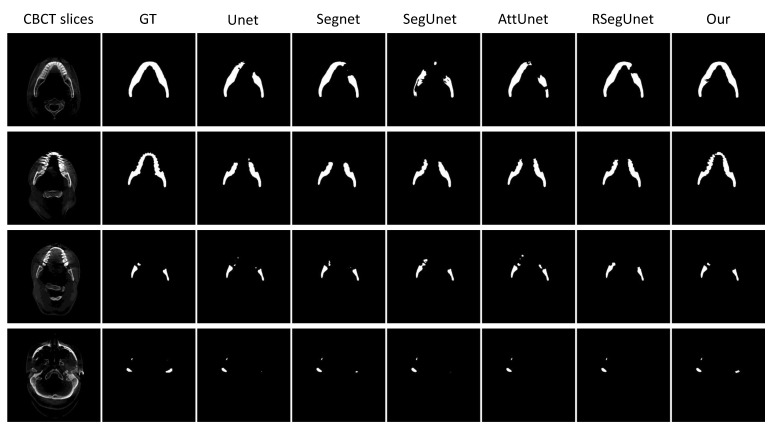
2D examples from CBCT dataset. From left to right: Original CT slice, Ground truth (GT), Unet, Segnet, SegUnet, AttUnet, RSegUnet, and the proposed method.

**Figure 5 jpm-11-00560-f005:**
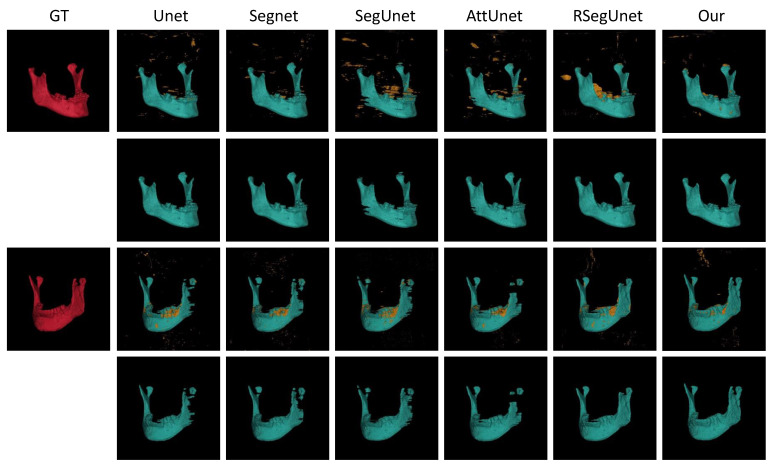
Visual 3D examples of final segmentations from the CBCT dataset. From left to right: Ground truth (GT), Unet, Segnet, SegUnet, AttUnet, RSegUnet, and the proposed method.

**Figure 6 jpm-11-00560-f006:**
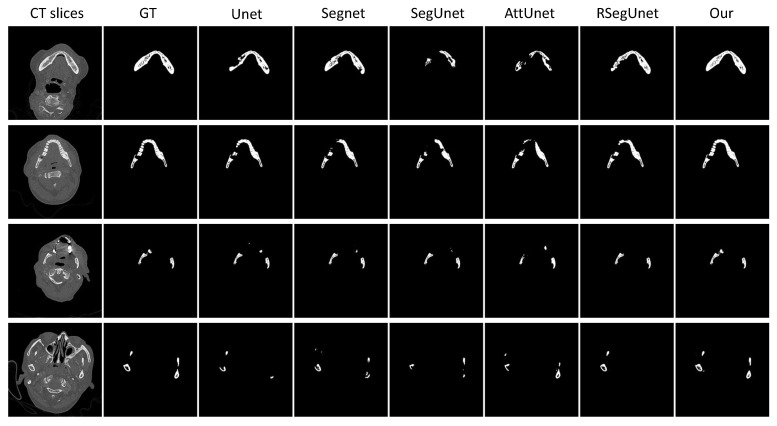
2D examples from CT dataset. From left to right: Original CT slice, Ground truth (GT), Unet, Segnet, SegUnet, AttUnet, RSegUnet, and the proposed method.

**Figure 7 jpm-11-00560-f007:**
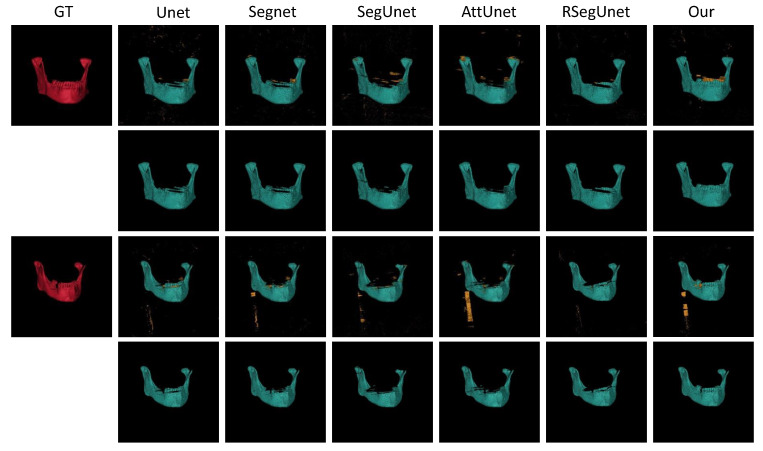
Visual 3D examples of final segmentations from the CT dataset. From left to right: Ground truth (GT), Unet, Segnet, SegUnet, AttUnet, RSegUnet, and the proposed method.

**Table 1 jpm-11-00560-t001:** Quantitative comparison of segmentation performance for the CBCT dataset between the proposed method and the state-of-the-art methods.

Methods	Dice (%)	$D_{\mathrm{ASD}}$ (mm)	$D_{95\mathrm{HD}}$ (mm)
Unet [18]	94.79 (±1.77)	2.0698 (±0.6137)	32.6401 (±22.0779)
SegNet [29]	94.93 (±1.74 )	1.7762 (±1.5937)	15.9851 (±26.5286)
SegUnet [15]	91.27 (±5.13)	3.1436 (±3.6049)	26.3569 (±34.9539)
AttUnet [30]	93.34 (±3.79)	3.9705 (±4.6460)	35.1859 (±42.3474)
RSegUnet [11]	92.26 (±5.66)	1.3133 (±0.7276)	7.2442 (±8.9275)
Ours	95.31 (±1.11)	1.2827 (±0.2780)	3.1258 (±3.2311)

**Table 2 jpm-11-00560-t002:** Quantitative comparison of segmentation performance for the CT dataset between the proposed method and the state-of-the-art methods.

Methods	Dice (%)	$D_{\mathrm{ASD}}$ (mm)	$D_{95\mathrm{HD}}$ (mm)
Unet [18]	87.61 (±5.13)	1.8779 (±0.7407)	9.2152 (±17.0825)
SegNet [29]	86.11 (±7.69)	1.6028 (±0.7194)	7.6235 (±15.1696)
SegUnet [15]	83.14 (±12.65)	2.4753 (±1.9507)	15.4372 (±25.1890)
AttUnet [30]	86.11 (±11.63)	1.6033 (±1.4386)	16.7041 (±24.2038)
RSegUnet [11]	86.48 (± 7.98)	1.3907 (± 0.7566 )	7.6591 (±16.7968 )
Ours	88.62 (±4.98)	1.2582 (±0.4102)	4.9668 (±5.0592)

**Table 3 jpm-11-00560-t003:** Quantitative comparison of segmentation performance for the PDDCA dataset between the proposed method and the state-of-the-art methods.

Methods	Dice (%)	$D_{\mathrm{ASD}}$ (mm)	$D_{95\mathrm{HD}}$ (mm)
Multi-atlas [31]	91.7 (±2.34)	-	2.4887 (±0.7610)
AAM [32]	92.67 (±1)	-	1.9767 (±0.5945)
ASM [33]	88.13 (±5.55)	-	2.832 (±1.1772)
CNN [8]	89.5 (±3.6)	-	-
NLGM [34]	93.08 (±2.36)	-	-
AnatomyNet [9]	92.51 (±2)	-	6.28 (±2.21)
FCNN [10]	92.07 (±1.15)	0.51 (±0.12)	2.01 (±0.83)
FCNN+SRM [10]	93.6 (±1.21)	0.371 (±0.11)	1.5 (±0.32)
CNN+BD [35]	94.6 (±0.7)	0.29 (±0.03)	-
HVR [36]	94.4 (± 1.3)	0.43 (± 0.12)	-
Cascade 3D Unet [37]	93 (±1.9)	-	1.26 (±0.5)
Multi-plana r [7]	93.28 (±1.44)	-	1.4333 (±0.5564)
Multi-view [38]	94.1 (±0.7)	0.28 (±0.14)	-
RSegUnet [11]	95.10 (±1.21)	0.1367 (±0.0382)	1.3560 (±0.4487)
SASeg [39]	95.29 (±1.16)	0.1353 (±0.0481)	1.3054 (±0.3195)
Our	94.57 (±1.21)	0.1252 (±0.0275)	1.1813 (±0.4028)

## Data Availability

For the Public Domain Database for Computational Anatomy Dataset (PDDCA) is available in https://www.imagenglab.com/newsite/pddca/ (accessed on 24 January 2019). Unfortunately, for reasons of ethics and patient confidentiality, we are not able to provide the sequencing data into a public database. The data underlying the results presented in the study are available from the corresponding author.

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
