# Peer review of "Mandible Segmentation of Dental CBCT Scans Affected by Metal Artifacts Using Coarse-to-Fine Learning Model"

_jpm, 2021, doi:10.3390/jpm11060560_

Round 1
Reviewer 1 Report
This is a well-written paper addressing an important issue of dental CBCTs. I feel their approach can help in developing the quality of dental radiographs in the future.
Reviewer 2 Report
Dear Authors
- Abstract is good but keywords should revise or carefully generate MeSH term.
- Line 38-55: Authors reported many studies but the flow of work is not well reported. Try to write in a good way.
- 3.1.1. CBCT dataset: this heading need careful attention. reference them.
- Before the conclusion, the heading tries to add the limitations of this work so research will process for further way.
- Double-check the references.
Reviewer 3 Report
The research is well designed and carried out.
Abstract: it is a good summary of the paper, and it is well organized.
Introduction contains enough background informations regarding the techniques involved and adequate references.
Figures and tables are adequate.
I will appreciate if you state in discussion section drawbacks and clinical application of your study.
